# Data mining methodology for response to hypertension symptomology— application to COVID-19-related pharmacovigilance

Xuan Xu[1,2,3], Jessica Kawakami[1,4,5], Nuwan Indika Millagaha Gedara[1,3,6], Jim E Riviere[1,7], Emma Meyer[1,4], Gerald J Wyckoff[1,4,5], Majid Jaberi-Douraki[1,2,3]*

[1]1DATA Consortium, www.1DATA.life, Olathe, United States; [2]Department of Mathematics, Kansas State University, Kansas City, United States; [3]Kansas State University Olathe, Olathe, United States; [4]School of Pharmacy, Division of Pharmacology and Pharmaceutical Sciences, University of Missouri-Kansas City, Kansas City, United States; [5]Molecular Biology and Biochemistry, School of Biological and Chemical Sciences, University of Missouri-Kansas City, Kansas City, United States; [6]Department of Business Economics, University of Colombo, Colombo, Sri Lanka; [7]Kansas State University and North Carolina State University, Kansas city, United States

*For correspondence: jaberi@k-state.edu

Competing interest: The authors declare that no competing interests exist.

## ABSTRACT

**Background:** Potential therapy and confounding factors including typical co-administered medications, patient's disease states, disease prevalence, patient demographics, medical histories, and reasons for prescribing a drug often are incomplete, conflicting, missing, or uncharacterized in spontaneous adverse drug event (ADE) reporting systems. These missing or incomplete features can affect and limit the application of quantitative methods in pharmacovigilance for meta-analyses of data during randomized clinical trials.

**Methods:** Data from patients with hypertension were retrieved and integrated from the FDA Adverse Event Reporting System; 134 antihypertensive drugs out of 1131 drugs were filtered and then evaluated using the empirical Bayes geometric mean (EBGM) of the posterior distribution to build ADE-drug profiles with an emphasis on the pulmonary ADEs. Afterward, the graphical least absolute shrinkage and selection operator (GLASSO) captured drug associations based on pulmonary ADEs by correcting hidden factors and confounder misclassification. Selected drugs were then compared using the Friedman test in drug classes and clusters obtained from GLASSO.

**Results:** Following multiple filtering stages to exclude insignificant and noise-driven reports, we found that drugs from antihypertensives agents, urologicals, and antithrombotic agents (macitentan, bosentan, epoprostenol, selexipag, sildenafil, tadalafil, and beraprost) form a similar class with a significantly higher incidence of pulmonary ADEs. Macitentan and bosentan were associated with 64% and 56% of pulmonary ADEs, respectively. Because these two medications are prescribed in diseases affecting pulmonary function and may be likely to emerge among the highest reported pulmonary ADEs, in fact, they serve to validate the methods utilized here. Conversely, doxazosin and rilmenidine were found to have the least pulmonary ADEs in selected drugs from hypertension patients. Nifedipine and candesartan were also found by signal detection methods to form a drug cluster, shown by several studies an effective combination of these drugs on lowering blood pressure and appeared an improved side effect profile in comparison with single-agent monotherapy.

**Conclusions:** We consider pulmonary ADE profiles in multiple long-standing groups of therapeutics including antihypertensive agents, antithrombotic agents, beta-blocking agents, calcium channel blockers, or agents acting on the renin-angiotensin system, in patients with hypertension associated with high risk for coronavirus disease 2019 (COVID-19). We found that several individual drugs have significant differences between their drug classes and compared to other drug classes. For instance, macitentan and bosentan from endothelin receptor antagonists show major concern while doxazosin and rilmenidine exhibited the least pulmonary ADEs compared to the outcomes of other drugs. Using techniques in this study, we assessed and confirmed the hypothesis that drugs from the same drug class could have very different pulmonary ADE profiles affecting outcomes in acute respiratory illness.

**Funding:** GJW and MJD accepted funding from BioNexus KC for funding on this project, but BioNexus KC had no direct role in this article.

## Editor's evaluation

The authors provide a comprehensive statistical analysis of anti hypertensive drug usage, including those use for pulmonary hypertension, in COVID-19 patients. Given the possible association between hypertension, use of blood pressure lowering medications, and COVID-19 risk, such data-driven analyses are important for drawing associative conclusions that could lead to future etiological experiments to identify specific causal mechanisms.

## Introduction

The coronavirus disease 2019 (COVID-19) pandemic continues with 115,094,614 confirmed cases and over 2.6 million deaths as of March 5, 2021 (*WHO, 2020*; *WHO, 2021*). Surprisingly, it is estimated that as high as 45% of infected individuals may remain asymptomatic, contributing to disease transmission and underlying the disparity in symptomology (*Oran and Topol, 2020*). A commonality of severe clinical course and mortality is comorbid conditions such as diabetes, heart disease, obesity, and hypertension (*CDC, 2019*). Hypertension was recognized early on as being a prevalent risk factor (*Zhou et al., 2020*), possibly due to its pervasiveness. Hypertension affects 23% of adults in China, where the original study was conducted, but affects 45% of US adults. Moreover, specific antihypertensive medications, namely angiotensin-converting enzyme inhibitors (ACEIs) and angiotensin-II receptor blockers (ARBs), target proteins of the renin-angiotensin system (RAS) (*James et al., 2014*). The RAS is intricately linked to initial infection and possibly the progression of COVID-19 through a RAS receptor, angiotensin-converting enzyme 2 (ACE2), which acts as the viral entry point of coronavirus SARS-CoV-2 (*Wiese et al., 2020*; *Li et al., 2003*).

In recent years, data science has emerged as a new and important discipline in medicine and healthcare. Different quantitative therapeutic efforts in drug repurposing or repositioning combined with adverse drug event (ADE) identification have led to more efficient therapies while improving the clinical course, lowering fatality, and decreasing cost burden (*Smith and Smith, 2021*). The previous work focused on the incidence of pulmonary ADEs associated with ACEI and ARB use in patients with hypertension and other comorbidities (*Stafford et al., 2020*; *Jaberi-Douraki et al., 2021*). The findings indicate that specific drugs—rather than entire classes—have higher incidences of pulmonary ADEs, which may have implications for treating patients diagnosed with COVID-19. Most epidemiological studies are not this granular as they do not analyze drug effects at the individual drug level but rather compare pharmacological classes. The current study examines additional drugs that more broadly target hypertension, including pulmonary hypertension, to describe methods used to identify clinically important patterns of ADE data. We utilized the Anatomical Therapeutic Chemical (ATC) classification system from the World Health Organization (WHO) Collaborating Center for Drug Statistics Methodology (https://www.whocc.no/). The ATC system classifies drugs based on site of action in addition to chemical, pharmacological, and therapeutic properties (*Rønning, 2001*). Here, we identify a clear signal distinct from different drugs in patients with hypertension as an underlying medical condition which helps to quantify the anomaly and unexpectedness of an ADE reported for a drug through disproportionality analysis. For this purpose, we proceeded with a specific pairwise analysis

of individual drugs compared to the drug classes using a modified empirical Bayes method to identify any distinctions between drugs within a class and compared to other classes.

In our previous work, 13 different pulmonary ADEs were selected based on clinical importance, and as they were prevalent among the top reported symptoms in patients with COVID-19, to assess the related variation due to adverse event differences (*Stafford et al., 2020*; *Jaberi-Douraki et al., 2021*). In the present work, we include 25 pulmonary, infectious diseases, or cardiac-associated ADEs. Our novel method identifies extraneous causes of differential reporting including sampling variance and selection biases by reducing the effect of covariates. This method is both adaptive (it removes different covariates for different drugs) and appropriate for the systematic application and routine analysis (*Tatonetti et al., 2012*). We hypothesize that drugs from the same class based on the ATC classification system could have different ADE profiles. For this purpose, penalized regression method will be used to detect clusters of drugs, may differ from the ATC classification, and will be validated by the Friedman test (*Evans et al., 2001*; *van Puijenbroek et al., 2002*; *Zorych et al., 2013*). Safety signals for a specific drug and associated adverse events are then identified and evaluated through different methods, such as the proportional reporting ratio (PRR) (*Evans et al., 2001*), the relative reporting ratio (RR) (*DuMouchel, 1999*), the information component (IC) (*Bate and Evans, 2009*), and the (EBGM) (*DuMouchel, 1999*). These methods are utilized to calculate the ratio of an ADE compared to the same event occurring with other drugs, however, PRR or RR is more liberal when an event incidence is small (*Duggirala et al., 2019*).

## Materials and methods

To derive the desired information from datasets, there are a few main methodological steps in this study. In the following, procedures are briefly illustrated in the workflow integrated by machine learning where some preprocessing points are first presented in *Figure 1*. This figure summarizes the steps in the preparation and analysis of the ADE database to make a decision and interpret the results; each step is detailed in the following subsections:

1. Working hypothesis: drugs from the same drug class could have different pulmonary ADE profiles affecting outcomes in acute respiratory illness, with potential implications in SARS-CoV-2 infection.
2. Designing error correction techniques for data scrubbing and retrieval.
3. Implementing data exploration technique for initial data analysis to visually explore and understand the characteristics of the data from post-marketing drug safety surveillance.
4. Data curation and annotation to organize and integrate data collected from various sources from the FDA, MedDRA, and ATC classification. This phase entails annotation, organization, clustering, and presentation of the assorted data types from the 1DATA databank.
5. ADE-associated information retrieval for patients with hypertension provides massive collections of reports to investigate ADE based on comparative population data analysis (1131 drugs and 1520 high-level terms (HLTs) corresponding to 480,236 spontaneous reports).
6. Integration of machine learning models (134 drugs were kept when used for more than 500 individual reports and EBGM were applied to assess 134 drugs).
7. Acquiring results after data preprocessing and cleansing significantly reduces the size of data and eliminates insignificant and noise-driven reports.
   I. 44 drugs were selected when EB05 >1 and the existence of two unique ADEs, 25 pulmonary HLTs were then filtered from 1152 HLTs;
   II. 22 drugs were selected by the GLASSO.
8. and 9. Enhancing decision and interpretation via data-driven machine learning to help identify incidences of pulmonary ADEs for potential therapy and confounding factors that may have implications for treating patients diagnosed with COVID-19, respectively.

### Data integration

The data were integrated into the 1DATA databank (www.1DATA.life) (*Xu et al., 2019*) from multiple sources, including the Food and Drug Administration (FDA) Adverse Drug Events Reporting System (FAERS), the Medical Dictionary for Regulatory Activities (MedDRA), and the ATC classification system. The FAERS database consists of voluntarily or mandatorily reported ADEs from healthcare professionals, manufacturers, and consumers; encompassing drug-related adverse occurrences pertaining

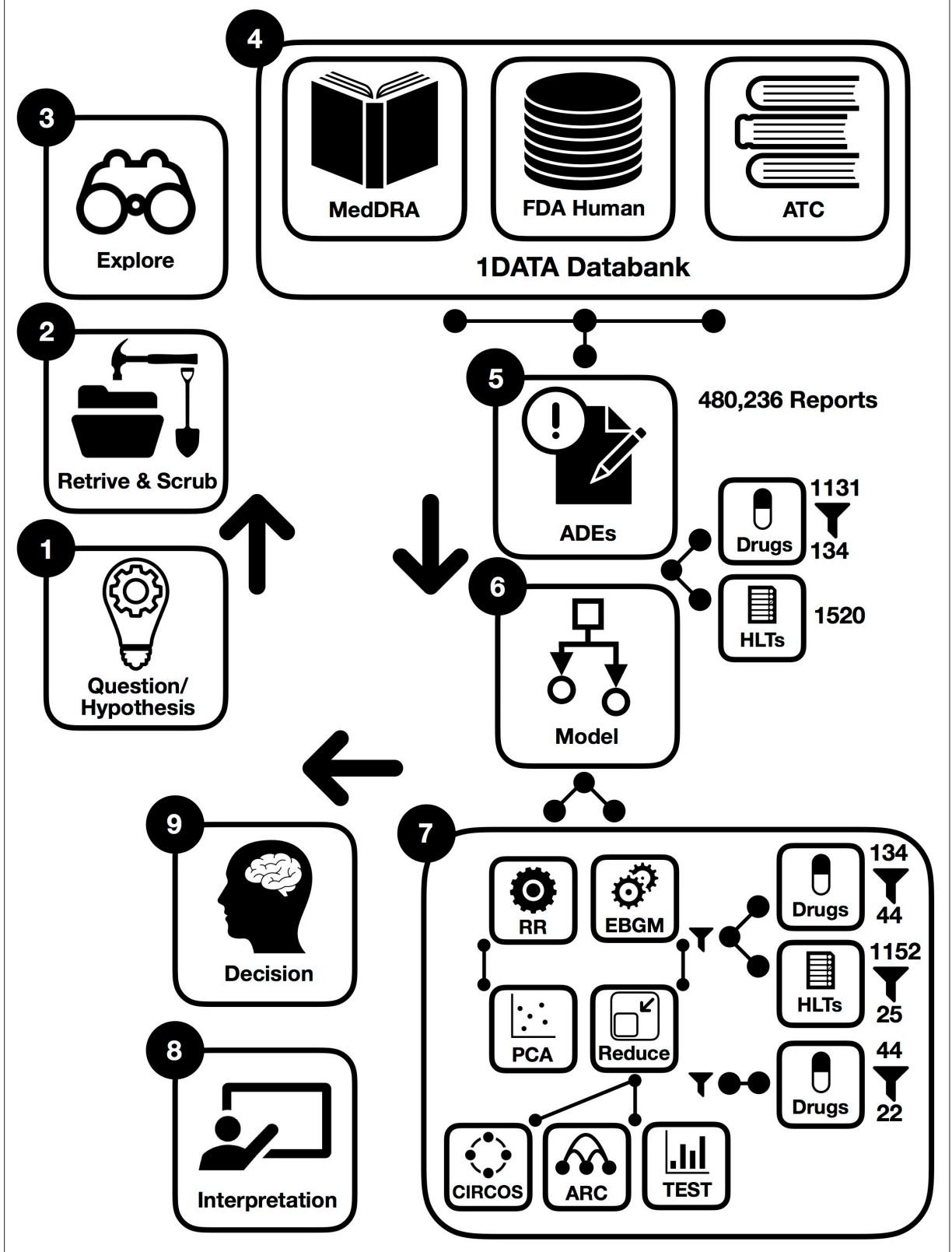

**Figure 1.** Workflow of data-driven methodology for pulmonary symptomology in hypertension using machine learning models from preprocessing and dictionary creation to storing tables in the database an analysis. As a part of data cleaning, we were also challenged by multiple technical issues when combining drugs: (i) there were many drugs' names that did not track a specific standard. (ii) Formulations of the same active ingredient with different generic or brand names for different routes of administration created confusion in collecting data (for instance, Revatio, Viagra, sildenafil, sildenafil

*Figure 1 continued*

citrate, APO sildenafil, sildenafil film-coated tablet, sildenafil citrate Aurobindo pharma, sildenafil Amneal Pharmaceuticals, Teva sildenafil, sildenafil Pfizer, sildenafil Greenstone, sildenafil Hormosan Filmtabletten, Revathio, sildenafil SUP, etc.). For this purpose, we combined drugs with or without salt, alcohol, etc. from different generic names and brand names.

to standard use, medical error, overdose, or product quality (*FDA Adverse Event Reporting System, 2014*). ADE reports from FAERS are typically coded in accordance with the preferred term (PT) level of MedDRA. The MedDRA provides an internationally recognized hierarchical terminology (system organ class [SOC], high-level group term (HLGT), HLT, PT, and lowest level term [LLT]) for coding ADE reports (*Mozzicato, 2012*). This study aggregates raw ADE reports to terms from the HLT and SOC levels. ATC classification is likewise an internationally applied hierarchical system for active drug substances based on site of action (organ or system) and mechanistic properties (therapeutic, pharmacological, and chemical). Drugs in this study were grouped according to ATC classification. Data integration into 1DATA occurred through the PostgreSQL 13.2 version (PostgreSQL Global Development Group), which allows concatenation of drug and ADE information (*Xu et al., 2019*; *PostgreSQL, 1996*).

## Adverse drug event

ADEs cause approximately 30 billion dollars a year of added healthcare expenses, along with negative—including fatal—health outcomes (*Xu et al., 2019*). The practice of prescribing drugs based on information from drug preapproval labeling may misrepresent or deprecate the incidence and prevalence of specific ADEs. The FDA defines the term 'adverse event' as: "any untoward medical occurrence associated with the use of a drug in humans, whether or not considered drug related, including the following: an adverse event occurring in the course of the use of a drug product in professional practice; an adverse event occurring from drug overdose whether accidental or intentional; an adverse event occurring from drug abuse; an adverse event occurring from drug withdrawal; and any failure of expected pharmacological action" (*FDA, 2020a*; *FDA, 2020b*).

## Relative risk

The main method used in this study, Bayesian shrinkage, is based on a baseline frequency, which is the relative risk or relative reporting ratio

$$RR_{ij} = \frac{N_{ij}}{E_{ij}}.$$

It compares a drug-ADE count, $N$, to its expected count, $E$. For instance, when $N_{ij}/E_{ij}$ is equal to 100, then $drug_i$ and $ADE_j$ occurred 100 times as frequently as the baseline frequency represents. A huge difference of occurrences between two drug-ADE pairs might lead to similar RR due to $E$ in the denominator, even statistically the same, but the frequency illustrates sampling variation. When more events of $ADE_j$ are caused by $drug_i$ higher than the same ADE in the database, $RR_{ij} > 1$. Drug-ADE surveillance should be triggered when large RR scores show up for specific drug-ADE pairs. However, the variability of RR for small counts drug-ADE pairs is unreliable, the high value of RR might be accidental.

## Principal component analysis

Principal component analysis (PCA) was obtained based on the log expected value of RR, log($E$), to analyze ADEs for different drugs, to reduce the features from the drug-ADE matrix. The distinct clusters from PCA plots were used to compare the similarities of drugs based on $E$. PCA was conducted using built-in function *PCA* in R (R 3.6.3 version, R Core Team, GNU GPL v2), and PCA biplots were produced using the R package *factoextra*, and 3D PC plots were produced using R package *plotly*.

## Gamma-Poisson shrinker

*DuMouchel, 1999*, proposed an empirical Bayes approach based on the Gamma-Poisson shrinker (GPS) algorithm to bring down the inflated value of *RR* due to small counts without impacting *RR* associated with large counts. Thus, the drug profile based on ADE could be reconstructed with reduced variation in *RR*. GPS redefines $RR_{ij}$ as $\lambda_{ij}=\mu_{ij}/E_{ij}$ drawn from a prior distribution with a mixture of two gamma distributions, $\mu_{ij}$ is the mean of the Poisson distribution of counts for $drug_i$ and $ADE_j$:

$$prior: \Pi\left(\lambda \vee \alpha_1, \beta_1, \alpha_2, \beta_2, P\right) = P \times gamma\left(\lambda \vee \alpha_1, \beta_1\right) + \left(1 - P\right) \times gamma\left(\lambda | \alpha_2, \beta_2\right)$$

which then gives the posterior probability from the components of the mixture model:

$$posterior: \lambda \vee N = n\ \Pi\left(\lambda \vee \alpha_1 + n, \beta_1 + E, \alpha_2 + n, \beta_2 + E, Q_n\right)$$

GPS shrinks RR scores by using EBGM from

$$EBGM = e^{E(log\lambda)}.$$

The shrinkage abates vagueness by reducing RR scores to a conservative level, which helps to alleviate false-positive signals, avoiding arbitrary drug-ADE assessment. The R package *openEBGM* was used to implement the GPS method (*Canida and Ihrie, 2017*).

## Correlation matrix and GLASSO

The profile of each drug comprises EBGM of all ADEs. The Pearson correlation matrix was constructed based on the EBGM between pairs of drugs. The vector

$$EB_i = \left(EB_{i1}, EB_{i2}, \ldots, EB_{ip}\right)$$

for $i \in \{1, 2, \ldots, n\}$ denotes the EBGM corresponding to $drug_i$. The Pearson correlation method determines the associations between pairwise vectors of reported drugs, which are the elements in the correlation matrix. This adjacency matrix was highly dense ($n \times n$), and it is difficult to graph the network when too many drugs (1131) are present. A penalized regression method, graphical least absolute shrinkage and selection operator (GLASSO), was then introduced to encourage sparsity in the adjacency matrix, in order to plot high-dimensional graphs from the correlation matrix (*Tibshirani, 1996*). An R package called *huge* was utilized to perform GLASSO (*Zhao et al., 2012*).

## Drug-ADE correlation diagram

The MedDRA hierarchy is multi-axial, for example, 'influenza' is from the PT level and is encompassed within two SOC levels 'respiratory, thoracic, and mediastinal disorders' and 'infections and infestations'. Therefore, the columns of EBGM calculations in the drug-ADE matrix involve HLTs from the 'respiratory, thoracic, and mediastinal disorders' and 'infections and infestations' levels. For better visualization, ADE columns of one drug were put in a block with other rows being zeros. The dimension of a drug-ADE matrix was expanded from ($m \times q$) to ($m \times mq$) where $m(< n)$, and $m = 22$ denotes the number of drugs selected by GLASSO from original n = 44 drugs, and $q = 17$ denotes selected ADEs described above.

## Reverse Cuthill-Mckee algorithm

Reverse Cuthill-McKee (RCM) is a bandwidth and profile reduction method, which permutes a sparse matrix into a band matrix with vertices reordered close to the diagonal (*Gibbs et al., 1976*). RCM in this study implemented in MATLAB R2019b (version 9.7; MathWorks Inc, Natick, MA; RRID:SCR_001622) was applied to arrange the connections between drugs and ADEs to encourage fewer crossings in Circos plots and arc diagrams. Circos plots and arc diagrams were generated using the R packages *edgebundleR*, *igraph*, and *ggraph* (*Bostock et al., 2016*).

## Friedman test

Using SAS (SAS University Edition version 9.4, Chapel Hill, NC), sample differences among antihypertensive drug groups according to therapeutic main group ATC (ACEIs, ARBs, BBAs, CCBs, and TDs) were evaluated for a pairwise comparison analysis with the assumption that data were not normally distributed using the non-parametric Friedman test for two independent unequal-sized data. The Friedman test was also applied to perform multiple comparison tests (p-value for statistical significance <0.05). The p-values in Table 4, Figures 4 and 5, when they are less than 0.05, indicate significant differences across ATC classes or GL Clusters. In addition, pairwise comparison analysis was completed in SAS in order to estimate how any two ATC classes differ as well as GL Clusters. The significance level of comparing drugs in ATC classes/GL Clusters against each other was adjusted

**Table 1.** Drug class after applying first the two filtering rules to obtain 44 drugs and then the elimination process from the penalized regression graphical least absolute shrinkage and selection operator (GLASSO) to obtain 22 drugs.

| Drug class | # Reports (Total 612,733) | # Drugs after initial filtering (total 134) | # Drugs correspond to ≥2 ADEs in HLT codes when EB05 > 1 (total 44) | Drugs using GLASSO (total 22) |
|---|---|---|---|---|
| ACEIs | 69,327 | 13 | 3 | 1 |
| ARBs | 87,415 | 8 | 5 | 3 |
| Other RAS agents | 3,471 | 1 | 0 | 0 |
| Other antihypertensive | 120,425 | 14 | 7 | 4 |
| Antithrombotic agents | 67,767 | 10 | 7 | 3 |
| Beta blocking agents | 74,574 | 13 | 3 | 1 |
| Calcium channel blockers | 86,399 | 18 | 10 | 6 |
| Diuretics | 29,394 | 14 | 3 | 1 |
| Lipid modifying agents | 2,634 | 4 | 0 | 0 |
| Urologicals | 18,186 | 4 | 2 | 2 |
| Vasoprotectives | 909 | 1 | 0 | 0 |
| Combinations | 52,232 | 34 | 4 | 1 |

using a rigorous paired Wilcoxon signed-rank test with Bonferroni correction to control family-wise type I error (*Eisinga et al., 2017*).

## Results

### Preprocessing and data cleaning

Here, we briefly explain the data preprocessing and cleansing that will be used in different subsections. The focus of each subsection is given by the amount of data that will be used. A total of 480,236 spontaneous ADE reports for patients with hypertension were retrieved from the 1DATA databank of the FAERS database from the first quarter of 2004 until the first quarter of 2020. Alternatively, ADEs can be categorized by drug for a total of 612,733 reports (*Table 1*) arising from patients taking more than one drug. For example, a single ADE reported for a patient taking two different drugs, will generate one ADE report for each drug. This hypertension dataset was aggregated to 1520 ADEs in HLT codes corresponding to 1131 drugs with unique active substances. Next, drugs were excluded when the number of ADEs due to the fact that each drug was reported less than 500 times, accounting for approximately less than 0.1% of the data. Furthermore, 98.8% of the data corresponded to 134 of the 1131 drugs (*Table 1* with the column header: # Drugs after initial filtering; this dataset will be exploited to calculate the relative risk for the disproportionality measures of a drug-ADE occurrence). This study focused on the 98.8% of the data remaining after the elimination of insignificant and noise-driven reports. The 134 drugs were grouped according to the following ATC drug classes (*Table 1*): ACEIs, ARBs, other RAS agents, other antihypertensives agents (AHAs), antithrombotic agents (ATAs), beta-blocking agents (BBAs), calcium channel blockers (CCBs), diuretics, lipid-modifying agents, urologicals (UAs), vasoprotectives, and combinations of antihypertensives (COMBs).

Since there were five unrelated pulmonary ADEs in the database (*coronavirus infections, conditions associated with abnormal gas exchange, neonatal hypoxic conditions, newborn respiratory disorders NEC, pulmonary hypertensions*), the hypertension dataset was further reduced to reports corresponding to the following 30 pulmonary ADEs, see *Supplementary file 1*. Of the 30 pulmonary ADEs, 5 ADEs were additionally excluded from the analysis since there were no reports regarding these ADEs (*Supplementary file 1*).

**Table 2.** The number of pulmonary adverse drug events (ADEs) when relative reporting ratio (RR) larger than two or the fifth quantile of empirical Bayes geometric mean (EBGM), EB05, larger than two after graphical least absolute shrinkage and selection operator (GLASSO) filtering process implemented in Table 1.

| Drug | # Pulmonary ADEs | Order by EBGM | # Pulmonary ADEs | Order by RR |
|---|---|---|---|---|
| Macitentan | 16 | 1 | 10 | 2 |
| Bosentan | 14 | 2 | 5 | 11 |
| Epoprostenol | 11 | 4 | 9 | 4 |
| Selexipag | 10 | 5 | 10 | 2 |
| Sildenafil | 10 | 6 | 7 | 6 |
| Tadalafil | 10 | 7 | 3 | 44 |
| Beraprost | 7 | 10 | 13 | 1 |
| Nifedipine | 5 | 13 | 5 | 11 |
| Candesartan | 4 | 16 | 3 | 34 |
| Althiazide/Spironolactone | 3 | 20 | 4 | 18 |
| Bisoprolol | 3 | 21 | #N/A | #N/A |
| Imidapril | 3 | 24 | 5 | 11 |
| Azelnidipine | 2 | 30 | 4 | 23 |
| Azilsartan Kamedoxomil | 2 | 31 | 3 | 32 |
| Bendroflumethiazide | 2 | 32 | 3 | 33 |
| Benidipine | 2 | 33 | 5 | 11 |
| Cilnidipine | 2 | 34 | 5 | 11 |
| Doxazosin | 2 | 36 | 3 | 36 |
| Lercanidipine | 2 | 39 | 1 | 90 |
| Nicardipine | 2 | 40 | 5 | 11 |
| Rilmenidine | 2 | 42 | #N/A | #N/A |
| Telmisartan | 2 | 43 | 4 | 30 |

## Relative risk

One of the frequentist methods, the RR, based on the disproportionality measures of a drug-ADE occurrence compared to other drug-event combinations was applied to evaluate the weighting of drugs. To start the first analysis, a large contingency table was constructed for the entire data from 134 selected drugs based on their frequencies with respect to all 1520 reported ADEs in HLT codes from MedDRA. An assumption was imposed that an ADE is selected when RR >2 for a specific drug to assess the drug disproportionality in pharmacovigilance data by observed-expected ratios prior to the EBGM analysis, a more conservative and accurate way of disproportionality evaluation. It is worth mentioning that several studies have reported RR >1.5 or 2, or a particular threshold larger than 1 to justify the association with more confidence, especially in the presence of additive noise with the unidentified distribution (*Curtis et al., 1992*; *Balkau et al., 1999*; *Richardson et al., 2004*). Taking into account only 25 pulmonary ADEs in HLT codes when RR >2, the top 22 drugs with their corresponding number of pulmonary ADEs were obtained (*Table 2*). The order from the number of pulmonary ADEs is arranged based on the EBGM results after GLASSO elimination and the clustering given in *Table 1* that will be explained below. RR is also utilized to calculate the baseline frequency for EBGM and to construct the PCA as explained below.

## Principle component analysis

RR calculated for the expected frequency of 25 pulmonary ADEs associated with 134 drugs prescribed to patients with hypertension was used to generate the matrix for the PCA plot. This helped illustrate how the loadings of pulmonary features could separate drugs in a 2D or 3D space (*Supplementary file 2*). *Figure 2A* shows 134 drugs in a 2D PCA panel following a V shape scatter plot, no clear separation can be intuitively observed. ADEs (blue text) are also superimposed on the graph to obtain the corresponding loadings, direction, and weights with regard to the drugs (square shape). Generally, two clusters of pulmonary issues, one in the direction of the X-axis and another in the Y-axis, played an important role in separating these drugs in the space of PC1 and PC2. Twelve different pulmonary ADEs in HLTS codes (breathing abnormalities, bronchospasm and obstruction, coughing and associated symptoms, lower respiratory tract infections NEC, lower respiratory tract inflammatory and immunologic conditions, lower respiratory tract signs and symptoms, parenchymal lung disorders NEC, pneumothorax, and pleural effusions NEC, pulmonary oedemas, pulmonary thrombotic and embolic conditions, respiratory failures (excl neonatal), and respiratory tract disorders NEC) exhibited similar impact by differentiating these drugs when projected to PC1 (X-axis), and seven pulmonary ADEs in HLTs codes (bronchial conditions NEC, fungal lower respiratory tract infections, pleural conditions NEC, pleural infections and inflammations, respiratory signs and symptoms NEC, respiratory syncytial viral infections, and vascular pulmonary disorders NEC) contributed the most when projected to PC2 (Y-axis). A detailed contribution of all pulmonary variables is given in *Supplementary file 2* and will be reviewed in the Discussion.

*Figure 2B* illustrates how the pulmonary ADEs are separated in a 3D space. The first, second, and third principal components, PC1, PC2, and PC3, explain more than 90% of the variation. Drugs from different branches in the 3D plot represent distinctive effects of pulmonary ADEs on the separation. This figure shows the optimal representation of three active variables in biplots acquired by PCA by diminishing the effect of supplementary variables that have no or little influence on the pulmonary ADEs. Consistent with the previous finding, quinapril and trandolapril in hypertensive patients have a notably higher incidence of pulmonary ADEs compared with its drug class as well as other classes, *Figure 2B* (see also *Figure 2—figure supplement 1*).

## Empirical Bayesian geometric mean

While the RR method is widely utilized due to its simplicity and user-friendly processing, it is difficult to dismiss high variability for infrequent occurrences. The assessment of drugs or ADEs based on RR is variable because of information that the RR methodology does not include, including underreported or overreported events. To assess the effect that the RR methodology has when a small number of ADE occurrences are compared to the whole database, the fifth percentiles from the lower confidence interval of EBGM (EB05) were used as a very conservative alternative, and the results are compared to RR. This assessment was performed using EBGM, is reported similar to the prevalence evaluation using RR values from above. The frequencies of a single drug having multiple ADEs in HLT groups or a single HLT ADE occurrence in multiple drugs were calculated. It was then found that the top 10 drugs with pulmonary ADEs consisted of AHAs, ATAs, and UAs. Bosentan, tadalafil, treprostinil, and beraprost based on EBGM were ranked substantially higher than their corresponding ranks when using RR, with respect to pulmonary ADEs. This suggests that the conservative, EBGM method with a fifth percentile cut-off will allow for the examination of large datasets of ADEs when high variability is present in the number of ADEs across drugs or drug classes, and still allow for a robust reporting methodology as compared to the RR methodology. This allows analysis of very large sets of drugs and ADEs (such as approximately 500,000×134 matrix here) without loss of sensitivity or imparting an over-emphasis on ADEs from infrequently prescribed drugs.

## GLASSO

The total number of distinct drugs used by patients with hypertension was 134 after filtering out drugs with very low frequency (<0.001) in the Principle component analysis section. EBGM data were used to construct the new feature matrix for different drug classes. Then 44 drugs were selected based on two conditions: (1) the lower confidence interval of EBGM, EB05, of drugs was larger than 1, and (2) a minimum of two different pulmonary ADEs is associated with each drug, (*Table 1*) . It was found that few drugs in ACEIs, diuretics, and combinations tended to cause pulmonary issues. More than half

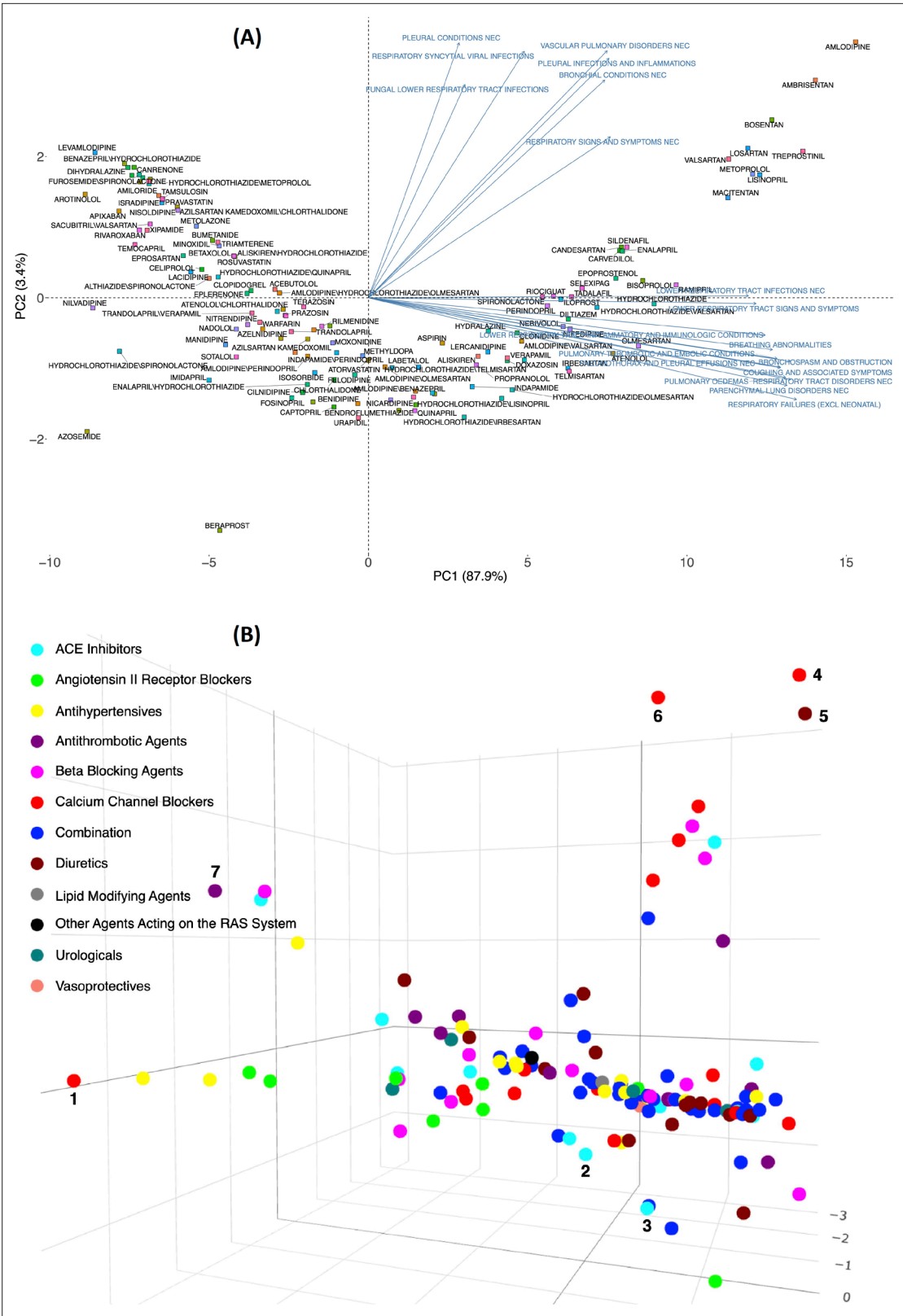

**Figure 2.** Principal component analysis of the expected count for 134 drugs (from 12 Anatomical Therapeutic Chemical [ATC] drug classes) in 2D (**A**) and 3D (**B**) spaces using the log expected value of RR, $logE$. In Panel B, individual drugs are (significantly) separated on the extreme edges are marked by (1), amlodipine, (2) quinapril, (3) trandolapril, (4) nilvadipine, (5) azosemide, (6) azelnidipine, and (7) treprostinil. An interactive figure can be found on the 1DATA home page. Click the following URL to see the figure: https://1data.life/pages/publication/figure1B.html.

*Figure 2 continued on next page*

*Figure 2 continued*

The online version of this article includes the following figure supplement(s) for figure 2:

**Figure supplement 1.** Principal component analysis of expected count for all the 117 drugs when excluding antihypertensive drugs in 2D (**A**) and 3D (**B**) space using empirical Bayes geometric mean (EBGM) for angiotensin-converting enzymes (ACEIs), angiotensin-II receptor blockers (ARBs), beta-blocking agents (BBAs), calcium channel blockers (CCBs), and TDs.

of the drugs were in ARBs, AHAs, ATAs, and CCBs when considering two different pulmonary ADEs at the HLT level. After two filtering steps, 44 drugs were set as the input for the penalized regression GLASSO. To have an adequate number of correlated drugs, the tuning parameter $\lambda$ of GLASSO was adjusted to shrink the less associated drugs to 0, which accounted for 50% of the selected drugs. The remaining 22 drugs selected by the GLASSO method based on Pearson correlation were classified using the therapeutic group Cardiovascular System (C01: Cardiac Therapy, C02: Antihypertensives, C03: Diuretics, C04: Peripheral Vasodilators, C05: Vasoprotectives, C07: Beta Blocking Agents, C08: Calcium Channel Blockers, and C10: Lipid Modifying Agents), except for agents acting on RAS, which are the pharmacological subgroup C09 (C09A: ACE Inhibitors, C09B: ACE Inhibitors, Combinations, C09C: Angiotensin II Receptor Blockers (ARBS), and C09X: Other Agents Acting on the RAS), the third level was applied to classify the RAS drugs since RAS drugs are the frontline agents in hypertension, *Table 1* and *Supplementary file 3*.

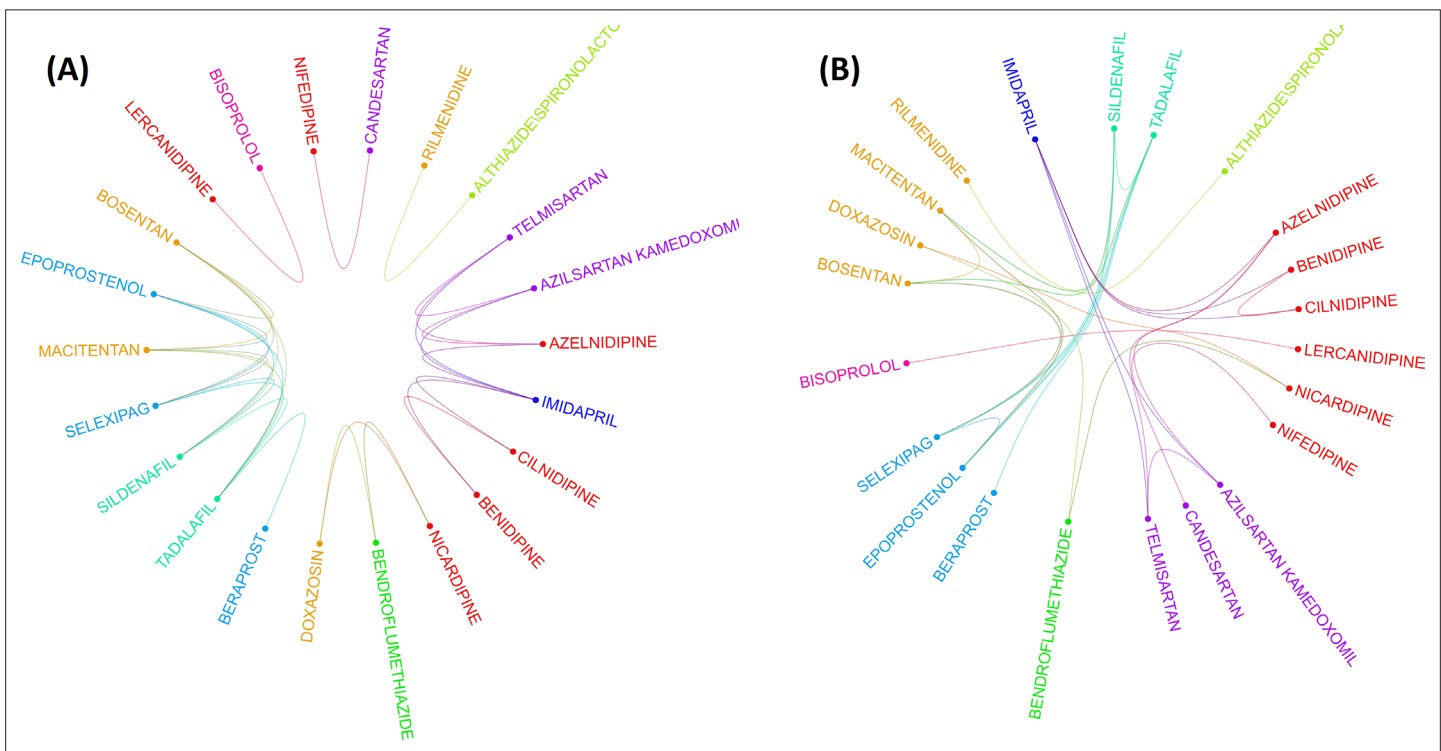

**Figure 3.** Two layouts of Circos plot for 22 hypertensive drugs selected by graphical least absolute shrinkage and selection operator (GLASSO). Circos plots of drugs were obtained based on the empirical Bayes geometric mean (EBGM) matrix after applying GLASSO. Edge bundling linkages for better visualization and drugs were selected by GLASSO with edge bundling. Grouped drugs based on their classes were assigned the same color based on their classes (**A**). Applying reverse Cuthill-McKee (RCM) reordering and edge bundling for grouping drugs based on the Anatomical Therapeutic Chemical (ATC) class and edge bundling (**B**).

The online version of this article includes the following figure supplement(s) for figure 3:

**Figure supplement 1.** Two layouts of Circos plot for 22 hypertensive drugs selected by graphical least absolute shrinkage and selection operator (GLASSO) when excluding antihypertensives agents (AHAs).

**Figure supplement 2.** Arc diagram visualization of 22 drugs selected from graphical least absolute shrinkage and selection operator (GLASSO) and associated pulmonary adverse drug event (ADE)-drug combination.

## Circos plot

The drug-drug correlation matrix with shrinkage is displayed in a circular layout, depicting drug class and associations between drugs from different classes (*Figure 3*). For drugs in ACEIs, ARBs, AHAs, and BBAs, no association was observed between drugs within the same class. More within-class associations were depicted in AHAs, CCBs, and combinations. *Figure 3A* shows the association between the remaining 22 drugs after then the elimination process from the penalized regression GLASSO. It means that the clustering was done after applying the GLASSO method which is a dimensionality reduction method. After these stringent filtering methods, drug classes exhibit very low significant correlations between drugs from the same class. This result is observed in *Figure 3A* by very few associations between drugs in the same class. Therefore, drug clustering using the RCM reordering method was employed in *Figure 3B*, with bridges connecting associated drugs. Without a doubt, this analysis corroborates the hypothesis that drugs from the same ATC class may have different pulmonary ADE profiles.

Given the 22 drugs selected by GLASSO, *Table 2* shows the assessment of drugs exclusively with respect to their pulmonary events. In the second column, # pulmonary ADEs defines the number of drug-ADE pairs from EBGM, which are depicted in the following section. Similarly, # pulmonary ADEs in the fourth column denote the results when RR is larger than 2. The order of drugs listed in *Table 2* is calculated based on the original 44 drugs from the EBGM scores and only the arrangement for the remaining 22 drugs out of 44 drugs is shown here. Beraprost showed 13 pulmonary ADE profiles reported more commonly than other drugs used for patients with hypertension based on the estimated RR. Macitentan and Selexipag were equally located in the second most commonly reported drugs, each of which with 10 pulmonary ADEs. In contrast, beraprost was corrected from being the top drug with most pulmonary issues and then ranked down to the tenth location by EBGM. The assessment for bosentan and tadalafil also changed radically when the comparative analysis was done using RR or EBGM.

From GLASSO and *Table 2*, the ADE profiles can be obtained in HLT groups for each drug in the newly identified group class, which we call GLASSO (GL) Clusters. The ADEs together with the drug classes from ATC and GL Clusters based on EB05 > 1 are arranged in *Table 3* and depicted by an arc diagram in *Figure 3—figure supplement 2* and *Supplementary file 4*. It is apparent from *Figure 3*, *Figure 3—figure supplement 1*, and *Table 3* that GL Cluster 1 consists of most associated drugs with most pulmonary ADEs assessed by EBGM (see also *Supplementary file 5*).

## Friedman test and multiple pairwise comparisons

To test the significant difference between drugs grouped by the original ATC classes and the GL Clusters, which were from a shrinkage correlation matrix, a non-parametric Friedman test was applied to compare separately the magnitude of difference when drugs in the same group for the ATC classes or the GL Clusters. *Table 4* summarizes the results of the p-value for different comparative analyses in the ATC classes or the GL Clusters. A p-value of 0.199 indicates that no differences in EBGM of pulmonary ADEs for different drugs in GL Cluster 1 when excluding tadalafil. Similarly, GL Clusters 2, 3, 4, 5, and 6 showed no significant differences in EBGM, respectively (*Table 4* and *Supplementary file 6*). However, given the original ATC class drugs belonging to, the Friedman test did show significant differences in six of the ATC classes before GLASSO. The same test was applied to 22 drugs selected from GLASSO, only drugs in UAs showed no significant differences in EBGM of pulmonary ADEs. This shows that instead of grouping drugs from the same ATC class, isolated groups from GLASSO showed homogeneity.

Pairwise drug class comparisons based on ATC class are shown for all the pairs (nine drug classes: ACEIs, ARBs, AHAs, ATAs, BBAs, CCBs, COMBs, TDAs, and UAs) in *Supplementary file 7*. The EBGM scores from the pulmonary ADE profiles were statistically significant for the nine ATC classes using the Friedman test (p-value = 0.0072, *Figure 4* and *Figure 4—figure supplement 1*). Pairwise comparisons showed no significant differences among any two ATC classes from the adjusted p-value (*Supplementary file 7*). However, using drug class determined by GLASSO, Wilcoxon signed-rank test between groups revealed significant differences in EBGM of pulmonary ADEs between GL Cluster 1 and GL Clusters 3, 4, and 5, respectively, compared to the pairwise comparisons between ATC groups, *Supplementary file 8*, *Figure 5*, and *Figure 5—figure supplement 1*. Drugs in GL group 1 showed significantly higher EBGM regarding pulmonary events. Friedman test confirming EBGM profile of selected drugs from GLASSO could be used for comparative analysis of drugs regarding certain indications.

**Table 3.** Comparative analysis of each drug and associated pulmonary adverse drug events (ADEs) based on the new classification from different graphical least absolute shrinkage and selection operator (GLASSO) (GL) Clusters.

| Drug | Drug class | ADEs for EB05 > 1 (n) * | GL Cluster |
|---|---|---|---|
| Macitentan | AHAs | 1–15,17 (16) | 1 |
| Bosentan | AHAs | 1,2,4–15 (14) | 1 |
| Epoprostenol | ATAs | 1,2,4–9,11,12,15 (11) | 1 |
| Selexipag | ATAs | 2,4–12 (10) | 1 |
| Sildenafil | UAs | 1,2,4–12 (10) | 1 |
| Tadalafil | UAs | 1,2,4–12 (10) | 1 |
| Beraprost | ATAs | 1,2,5–9 (7) | 1 |
| Nifedipine | CCBs | 1–3,15,16 (5) | 2 |
| Candesartan | ARBs | 1,3,14,16 (4) | 2 |
| Althiazide\Spironolactone | COMBs | 4,10,11 (3) | 3 |
| Rilmenidine | AHAs | 4,10 (2) | 3 |
| Bisoprolol | BBAs | 1,2,14 (3) | 4 |
| Lercanidipine | CCBs | 1,14 (2) | 4 |
| Imidapril | ACEs | 1–3 (3) | 5 |
| Azelnidipine | CCBs | 1,3 (2) | 5 |
| Azilsartan Kamedoxomil | ARBs | 1,3 (2) | 5 |
| Benidipine | CCBs | 1,2 (2) | 5 |
| Cilnidipine | CCBs | 1,2 (2) | 5 |
| Telmisartan | ARBs | 1,3 (2) | 5 |
| Bendroflumethiazide | TDAs | 3,13 (2) | 6 |
| Doxazosin | AHAs | 3,13 (2) | 6 |
| Nicardipine | CCBs | 3,13 (2) | 6 |

*Below ADEs can be found corresponding to each drug:
1. Parenchymal lung disorders NEC.
2. Pneumothorax and pleural effusions NEC.
3. Lower respiratory tract inflammatory and immunologic conditions.
4. Respiratory tract disorders NEC.
5. Breathing abnormalities.
6. Lower respiratory tract signs and symptoms.
7. Pulmonary oedemas.
8. Respiratory failures (Excl Neonatal).
9. Vascular pulmonary disorders NEC.
10. Bronchospasm and obstruction.
11. Coughing and associated symptoms.
12. Respiratory syncytial viral infections.
13. Bronchial conditions NEC.
14. Pulmonary thrombotic and embolic conditions.
15. Lower respiratory tract infections NEC.
16. Fungal lower respiratory tract infections.
17. Pleural infections and inflammations.

**Table 4.** The Friedman test for drugs in Anatomical Therapeutic Chemical (ATC) class and graphical least absolute shrinkage and selection operator (GLASSO) class.

| ATC class | p-value (44 drugs) | p-value (22 drugs) | GL Cluster | The p-value for 22 drugs |
|---|---|---|---|---|
| ACEIs | 0.271 | – | 1 | **<0.001** (0.199, when excluding tadalafil) |
| ARBs | **<0.001** | **<0.001** | 2 | 0.110 |
| AHAs | **<0.001** | **<0.001** | 3 | 0.884 |
| ATAs | **<0.001** | **<0.001** | 4 | 0.346 |
| BBAs | **0.0232** | – | 5 | 0.127 |
| CCBs | **0.001** | **0.001** | 6 | 0.0522 |
| COMBs | 0.236 | – | | |
| TDAs | **0.0329** | – | | |
| UAs | 0.127 | 0.127 | | |

The p-value for statistical significance is <0.05.

## Discussion

The future of large-scale biomedical science is data-driven decision-making and AI knowledge-based development and validation. AI-enabled technologies can help in better understanding disease indication occurrence and disease determinants or patterns. Quantitative methods have countlessly been applied in various medical fields of study, for example, measurement of disease frequency, prevalence or incidence; evaluation of source of bias and variation of observational studies; multivariate data analysis of risk factors such as applied logistic regression analysis; machine learning for survival analysis or analysis of time at risk (survival) data; boosting power for clinical trials using AI-assisted analysis, etc. In this study, we aimed to apply AI-driven methodologies involving EBGM and GLASSO techniques in predicting SARS-Cov-2 comorbidity for high-risk populations with hypertension.

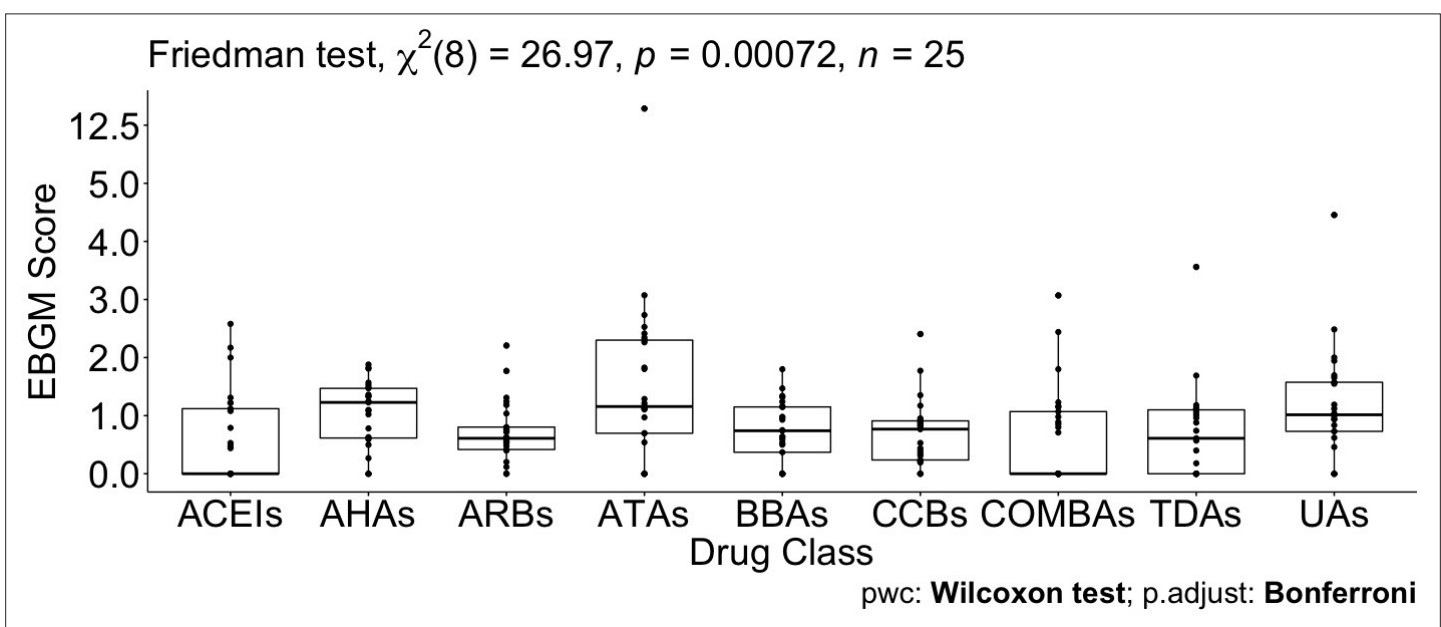

**Figure 4.** Pairwise Wilcoxon signed-rank test between different Anatomical Therapeutic Chemical (ATC) classes. No pairwise significant comparison was found similar to *Supplementary file 7*. But the group comparison was highly significant, p-value = 0.00072.

The online version of this article includes the following figure supplement(s) for figure 4:

**Figure supplement 1.** Pairwise Wilcoxon signed-rank test between different Anatomical Therapeutic Chemical (ATC) classes.

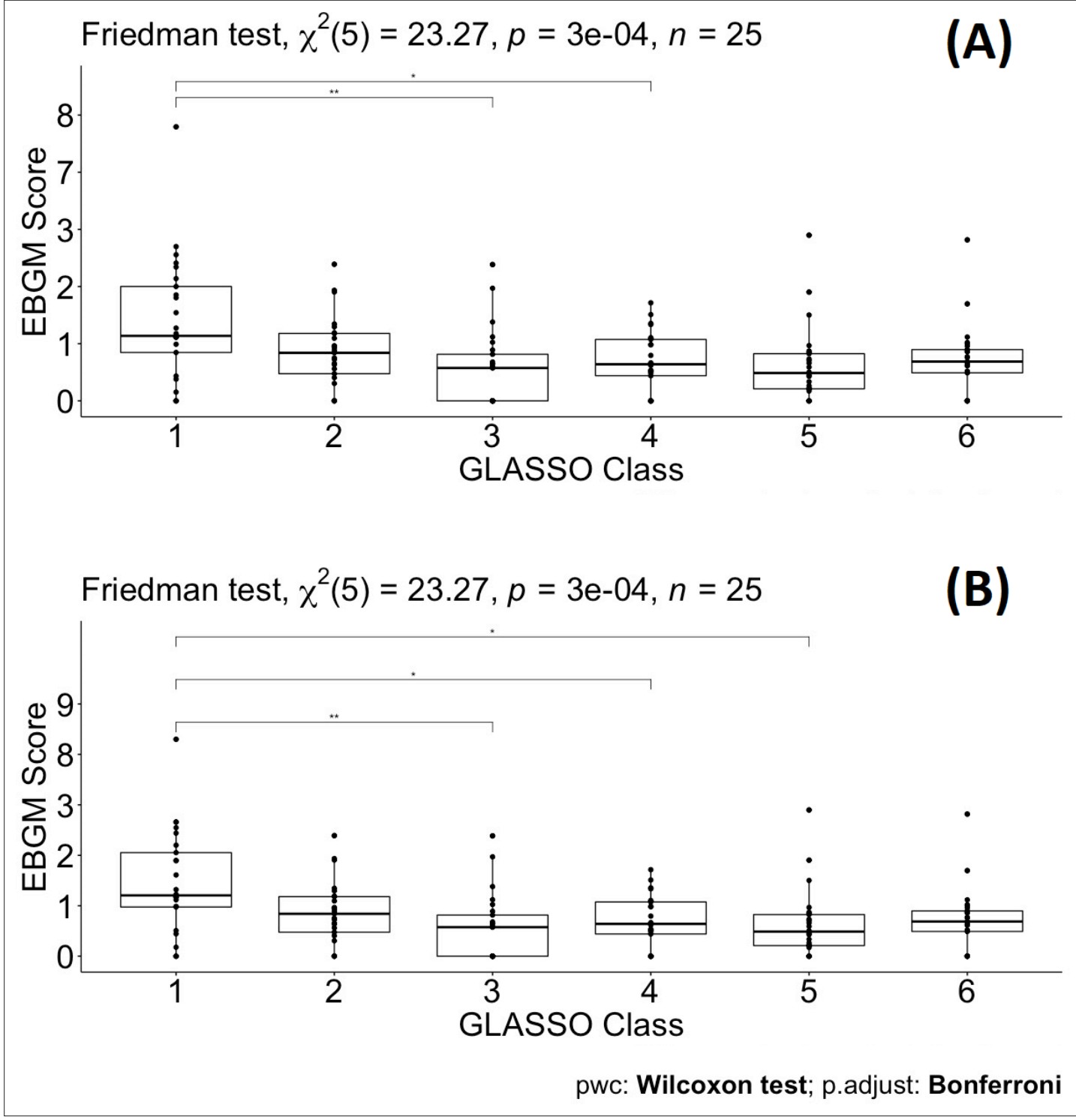

**Figure 5.** Pairwise Wilcoxon signed-rank test between different classes defined by graphical least absolute shrinkage and selection operator (GLASSO) (**A**) and pairwise Wilcoxon signed-rank test between different classes defined by GLASSO excluding tadalafil (**B**).

The online version of this article includes the following figure supplement(s) for figure 5:

**Figure supplement 1.** Pairwise Wilcoxon signed-rank test for different classes defined by graphical least absolute shrinkage and selection operator (GLASSO) (**A**) and the same test for different classes defined by GLASSO excluding warfarin (**B**) similar to *Supplementary file 8*.

Quantitative methods, that is, PPR, RR, ROR, EBGM have been used to detect signals for spontaneously reported data. After filtering data by quantitative methods, we proposed that selected drug-ADE based on drug association mechanism would be a valuable procedure for clinical review and comparison of similar drugs with similar ADE profiles. In this study, we demonstrated a systematic way of filtering and selecting data that addresses the noise inherent to such data. None of these methods are free from including false-positive and false-negative signals, however, EBGM and the IC are recommended over other quantitative methods when evaluating by mean average precision (*Zorych et al., 2013*). This helped the workflow to build a model to understand the bias-variance tradeoff to achieve a balance between the two desirable but incompatible features. Given the absence of a gold standard, no available method is overwhelmingly better than the others (*Bate and Evans, 2009*). The confirmatory methods proposed in this study (GLASSO and Friedman test) for assessing quantitative methods could reveal the strengths and drawbacks of the methods.

Drugs from different branches in the 3D plot represent distinctive effects of pulmonary ADEs on the separation. For example, PC3 is dominated by fungal, PC2 by more pleural and vascular, and PC1 by respiratory tract effects (see *Supplementary file 2*). PCs were constructed using the expected counts of a drug and a pulmonary ADE through a linear combination. The spatial separation of drugs indicated that drugs at the perimeter of each branch (numbered) performed disparately regarding pulmonary ADE profiles, suggesting they may not best be managed as having ADE profiles defined by their class. This figure shows the optimal representation of three active variables in biplots acquired by PCA by diminishing the effect of supplementary variables that have no or little influence on the pulmonary ADEs. Using the Friedman test, these separated drugs have significant differences between their drug classes and compared to other drug classes.

The consistency of the Friedman test and GLASSO to capture EBGM signals of drugs used in small and large populations could be a beneficial tool for drug comparative analysis. *Xu et al., 2019*, and *Stafford et al., 2020*, have already applied two methods in pharmacovigilance to animal and human data separately. This study proposed and successfully combined penalized regression together with the non-parametric Friedman test in considering to a better visualization of drug-drug and drug-ADE associations. The RR method is widely utilized due to its simplicity and user-friendly processing. RR, however, may be highly variable for small occurrences of an event. The assessment of drugs or ADEs based on RR showed unstable performance, especially for hidden information. The estimates of small occurrences compared to the whole database were also inflated for events. To correct these issues, the fifth percentiles from the lower confidence interval of EBGM (EB05) was introduced to use as a conservative alternative compared to RR.

EBGM detected that 16 out of 25 pulmonary ADEs in MedDRA databases were associated with macitentan, followed by bosentan with 14 pulmonary ADEs. Both of these drugs belong to the endothelin receptor antagonist class of drugs and are utilized in pulmonary arterial hypertension to prevent vasoconstriction, fibrosis, and inflammation on vascular endothelium and smooth muscle (*Lexicomp, 2016*). Both drugs are proposed to curb the pulmonary vascular resistance to prevent right heart failure and death, however, pulmonary ADEs of both drugs can be of major concern compared to the outcomes of several other AHAs utilized in this study. At the same time, because these two medications are used in a disease affecting pulmonary function and commonly reported ADEs to include therapeutic failure, these drugs were not surprising to emerge among the highest with reported pulmonary ADEs and, in fact, they serve to validate the methods utilized in this paper. Conversely, doxazosin and rilmenidine were found to have the least pulmonary ADEs in selected drugs from hypertension patients since only two ADE signals were detected based on EBGM. Although it can be used in hypertension, doxazosin is primarily utilized for men with benign prostatic hyperplasia and works by blocking alpha-adrenergic receptors in the vascular smooth muscle, resulting in vasodilation (*Lepor et al., 1997*). Additionally, studies in countries outside of the US suggest that rilmenidine, a sympatholytic, has a favorable ADE profile for patients with hypertension and diabetes, it is not approved in the US (*Meredith and Reid, 2004*). After excluding GL Cluster 1, almost the same results can be observed for the remaining GL Clusters. It is also worth mentioning here that the results are shown in *Supplementary files 3; 5–8*, as well as *Figure 2—figure supplement 1*, *Figure 3—figure supplement 1*.

The second group found by EBGM and GL clustering consisted of two drugs from CCBs (nifedipine) and ARBs (candesartan) grouped in combination (*Figure 3*) and showed four similar pulmonary

ADEs: parenchymal lung disorders NEC, pneumothorax and pleural effusions NEC, lower respiratory tract inflammatory and immunologic conditions, and fungal lower respiratory tract infections. Several studies based on these drugs showed effective combination and blood pressure lowering effects in patients with hypertension and appeared an improved side effect profile in comparison with single-agent monotherapy (*Hasebe et al., 2005*; *Kjeldsen et al., 2014*; *Mancia et al., 2017*; *Fujikawa et al., 2005*). This is undoubtedly an interesting finding resulted from the EBGM analysis and demonstrated how these two drugs can be combined and investigated for pharmacokinetic assessment in drug development including bioavailability and bioequivalence, drug safety pharmacovigilance, and efficacy and comparative tolerability of the combination of nifedipine and candesartan (*Patterson and Jones, 2017*; *Midha and McKay, 2009*).

Our previous work showed that quinapril and trandolapril were significantly different from other ACEI and ARB drug classes (*Jaberi-Douraki et al., 2021*). Separating from its drug class was initially observed in *Figure 2* when the PCA biplot was performed. However, these two drugs will not be present when more precautionary methods are applied for several reasons: (1) The dataset is no longer the same as before which contain only ACEIs or ARB. (2) The methods are very differen. (3) Several other drugs and ADEs are added to the study, 134 as opposed to only 16 drug.(4) In the previous work, the focus was on analyzing 13 pulmonary ADEs at the PT level; however, in the current study, we obtained and compare 25 ADEs in HLT groups and each HLT contains several PT ADEs. To be more accurate, ADEs for the 25 ADEs in HLT groups contain approximately 200 different PT vs. only 13 ADEs. (5) The whole purpose of this study was to use EBGM as a much more accurate method compared to RR and RR estimation is also better than the PRR method used before. (6) The implementation of the filtering process of penalized regression GLASSO helps eliminate the insignificant and noise-driven reports.

Two drugs, tadalafil and sildenafil, are also used for the modulation of dopaminergic pathways and modifying risk factors to prevent and treat erectile dysfunction. Using the 1DATA database when curating the data for the medicinal products of these drugs and checking their active ingredients of tadalafil and sildenafil, the top products are found to be Adcirca (n = 32446) and Revatio (n = 21,358) marketed for the treatment of pulmonary arterial hypertension, respectively, and Cialis (n = 15,623) and Viagra (n = 20820) marketed to treat erectile dysfunction, respectively. We also assessed whether these drugs only show up at high doses or not. This also confirmed that the dose has an insignificant effect on the outcome of ADEs, data are given in *Supplementary file 9*.

As part of the future work, it is worth mentioning that this study aimed to reveal the potential risk of patients using hypertensive drugs in terms of pulmonary issues. The 1DATA database will be updated with MedDRA 24.0 that contains the new COVID-19 terms due to its outbreak. It has encouraged the authors to involve terms related to viral infections that facilitate the capture of ADEs caused by COVID-19 in patients with hypertension in the near future. In addition, the pulmonary ADEs of HLT codes in this study were filtered by setting the highest level, SOC, with the focus on respiratory, thoracic, and mediastinal disorders (n = 28), and infection class containing viral infection (n = 2). The plan is to include ADEs from the class of blood and lymphatic system disorders such as thrombosis, coagulation, or platelet disorders. In the big data era, as the spontaneous reports from different data sources including the FDA FAERS database (*FDA Adverse Event Reporting System, 2014*), the Vaccine Adverse Event Reporting System (VAERS) (*Chen et al., 1994*; *Shimabukuro et al., 2015*), and the WHO International Database are increasing in size; drug profiles-based ADEs can be established based on quantitative methods, retrieving the signals, or detecting new signals in large numbers of reports by different methods with the combination of clinical review is needed for pharmacovigilance.

## Additional information

### Funding

| Funder | Grant reference number | Author |
| --- | --- | --- |
| BioNexus KC | 20-7 | Gerald J Wyckoff<br>Majid Jaberi-Douraki |

| Funder | Grant reference number | Author |
|--------|------------------------|--------|

The funders had no role in study design, data collection and interpretation, or the decision to submit the work for publication.

## Author contributions

Xuan Xu, Data curation, Formal analysis, Methodology, Software, Validation, Visualization, Writing – original draft; Jessica Kawakami, Conceptualization, Investigation, Validation, Writing – original draft; Nuwan Indika Millagaha Gedara, Data curation, Methodology, Software, Validation, Visualization, Writing – review and editing; Jim E Riviere, Conceptualization, Investigation, Project administration, Writing – review and editing; Emma Meyer, Conceptualization, Investigation, Writing – review and editing; Gerald J Wyckoff, Conceptualization, Funding acquisition, Investigation, Methodology, Writing – review and editing; Majid Jaberi-Douraki, Conceptualization, Data curation, Funding acquisition, Investigation, Methodology, Project administration, Resources, Software, Supervision, Validation, Visualization, Writing – original draft, Writing – review and editing

## Author ORCIDs

Xuan Xu (ID) http://orcid.org/0000-0003-3349-7183
Jessica Kawakami (ID) http://orcid.org/0000-0002-7141-5698
Majid Jaberi-Douraki (ID) http://orcid.org/0000-0002-8505-6550

## Decision letter and Author response

Decision letter https://doi.org/10.7554/eLife.70734.sa1
Author response https://doi.org/10.7554/eLife.70734.sa2

# Additional files

## Supplementary files

• Supplementary file 1. 30 pulmonary ADEs.

• Supplementary file 2. Contribution of Pulmonary ADEs in 2D and 3D PCAs.

• Supplementary file 3. Frequency of pulmonary ADEs when RR larger than two or the 5th quantile of EBGM, EB05, large than two.

• Supplementary file 4. Description of arc diagram visualization.

• Supplementary file 5. Comparative analysis of drug and associated pulmonary ADEs in different GLASSO Clusters.

• Supplementary file 6. Friedman test for drugs in ATC class and GLASSO class.

• Supplementary file 7. A. Multiple comparisons of different ATC classes together with the adjusted p-value using the rigorous paired Wilcoxon signed-rank test with Bonferroni correction. B. Multiple comparisons of different ATC classes excluding AHAs and UAs.

• Supplementary file 8. A. Multiple comparisons of drugs from GL Clusters and multiple comparisons of drugs from GL clusters excluding Tadalafil. B. Multiple comparisons of drugs from GLASSO clusters and multiple comparisons of drugs from GLASSO clusters excluding Warfarin.

• Supplementary file 9. Dose distribution related to tadalafil and sildenafil and ADE.

• Transparent reporting form

• Source data 1. This folder contains the data for R programs in the Source Code files.

• Source code 1. This folder contains R codes for the manuscript: Data-Driven Methodology COVID19 Related Pharmacovigilance.

## Data availability

The source code and data used to produce results and analyses presented in this manuscript are available at: https://1data.life/pages/publication/data_driven_methodology_COVID19_related_pharmacovigilance/.

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
