## [Editor Report]

The authors provide a comprehensive statistical analysis of anti hypertensive drug usage, including those use for pulmonary hypertension, in COVID-19 patients. Given the possible association between hypertension, use of blood pressure lowering medications, and COVID-19 risk, such data-driven analyses are important for drawing associative conclusions that could lead to future etiological experiments to identify specific causal mechanisms.

---

## [Decision Letter]

**Decision letter after peer review:**

Thank you for submitting your article "Data-driven methodology for discovery and response to pulmonary symptomology in hypertension through AI and machine learning: Application to COVID-19 related pharmacovigilance" for consideration by *eLife*. Your article has been reviewed by 2 peer reviewers, and the evaluation has been overseen by a Reviewing Editor and Paul Noble as the Senior Editor. The reviewers have opted to remain anonymous.

Essential revisions:

1. Where possible, please elaborate on thresholds used for analysis in data processing and filtering.

2. Be sure to carefully state where multiple comparison corrections to the p-value were made and how. This is done in some parts but could be clearer in others.

3. Please clarify if clustering was done following dimensionality reduction or whether these were separate steps.

4. The use of buzz words like "machine learning" and "AI" is over done given the preponderance of work is based in traditional statistics. I would advise removing machine learning and AI from the title as it sort builds up an appearance that does not hold when read by a machine learning audience. A "data driven approach" is appropriate and best represents the work.

5. Where possible, remove first person in Results and Discussion.

6. Title: "discovery and response to pulmonary symptomology in hypertension". It is confusing (at least) to me what the authors want to say. For example, does "discovery" here refer to "diagnosis"?

7. Abstract: The one sentence in Methods is not clear and specific enough. In addition, it is hard to follow how the results derive to the conclusions.

8. Line 51: typo: indidvual -> individual

9. Lines 130-168: I would suggest to move the lists of the ADEs names (e.g., lines 152-165) into a table or a supplementary material.

10 Line 176: Please add a reference supporting the cut-off criterion (RR >2).

11. Figure 1: It is hard to follow how to interpret the results. Also, please increase the font size in the figure.

Joint Review:

The work presented uses a comprehensive, traditional statistical approach combined with some sub-elements of machine learning to identify associative patterns of anti-hypertensive drug usage among COVID-19 patients. The goal was to assess SARS-CoV-2 comorbidity among high-risk populations with hypertension. The statistical approach is fundamentally rigorous, although it does contain a series of subjective thresholding decisions (no domain context or specific statistical reason for the decision), particularly in in data processing and filter, that could potentially alter results, particularly for the lower use drug cases. The inability to justify each decision point is not uncommon in big data analysis. Nonetheless, the technical analysis and interpretation of results for the more frequently occurring drugs is robust, supported by statistical analysis, and provides important insights for the planning of future experiments further assessing the etiological mechanisms by which hypertension and/or anti-hypertensive treatment alters the risk and outcome of patients with SARS-CoV-2.

---

## [Author Response]

Essential revisions:1. Where possible, please elaborate on thresholds used for analysis in data processing and filtering.

Thank you for your comment. We have tried to address this comment in the Preprocessing and Data Cleaning section of Results and the first part of the Methods section which shows how we performed the data cleaning and apply the thresholds. We have added the explanation in the Method part (lines 142 – 163). Here are some more details about data processing. Briefly, 480,236 spontaneous reports were originally extracted from our database (1131 drugs and 1520 ADEs in HLT).

a) The first threshold was applied to eliminate reports when the corresponding drugs were reported less than 500 times. The number of drugs was reduced from 1131 to 134. This process only reduced the size of the dataset by 0.1% and we were left by over %99.9 of the data.

b) Relative risks and EBGM scores were then calculated for 134 drugs. PCAs were performed conditioning on 25 selected pulmonary ADEs.

c) EB05 > 1 and two different pulmonary ADEs were applied to obtain 44 drugs from 134.

d) The GLASSO method was then implemented to 44 drugs with pulmonary ADEs to select 22 drugs for the Circos plot and Freidman’s test.

Above are in the manuscript. The first paragraph of the Results has also explained how we applied the thresholds for the main steps of data processing. Below is the edited part in the Method section (lines 142 – 163):

1. Working hypothesis: drugs from the same drug class could have different pulmonary ADE profiles affecting outcomes in acute respiratory illness, with potential implications in SARS-CoV-2 infection.

2. Designing error correction techniques for data scrubbing and retrieval.

3. Implementing data exploration technique for initial data analysis to visually explore and understand the characteristics of the data from post-marketing drug safety surveillance.

4. Data curation and annotation to organize and integrate data collected from various sources from the FDA, MedDRA, and ATC classification. This phase entails annotation, organization, clustering, and presentation of the assorted data types from the 1DATA databank.

5. ADE-associated information retrieval for patients with hypertension provides massive collections of reports to investigate adverse drug events based on comparative population data analysis. (1131 drugs and 1520 HLTs corresponding to 480,236 spontaneous reports).

6. Integration of machine learning models. (134 drugs were kept when used for more than 500 individual reports and EBGM were applied to assess 134 drugs).

7. Acquiring results after data preprocessing and cleansing significantly reduces the size of data and eliminates insignificant and noise-driven reports.

I. 44 drugs were selected when EB05 >1 and the existence of two unique ADEs, 25 pulmonary HLTs were then filtered from 1152 HLTs;

II. 22 drugs were selected by the GLASSO.

8 and 9. Enhancing decision and interpretation via data-driven machine learning to help identify incidences of pulmonary ADEs for potential therapy and confounding factors that may have implications for treating patients diagnosed with COVID-19, respectively.

2. Be sure to carefully state where multiple comparison corrections to the p-value were made and how. This is done in some parts but could be clearer in others.

Thank you for pointing out this issue. The p-value was referred to the situation when we tested the significant difference among all the classes or clusters for pairwise and group comparisons. The adjusted p-value was specific for the pairwise comparison, for which we employed the Bonferroni correction. Some additional statements have been added in the Method part (Lines 275 – 285). Below is the edited part of the Method section:

“Using SAS (SAS University Edition version 9.4, North Carolina, U.S), sample differences among antihypertensive drug groups according to therapeutic main group ATC (ACEIs, ARBs, BBAs, CCBs, and TDs) were evaluated for a pairwise comparison analysis with the assumption that data were not normally distributed using the non-parametric Friedman test for two independent unequal-sized data. The Friedman test was also applied to perform multiple comparison tests (P-value for statistical significance < 0.05). The p-values in Table 4, Figure 4, and Figure 5, when they are less than 0.05, indicate significant differences across ATC classes or GL clusters. In addition, pairwise comparison analysis was completed in SAS in order to estimate how any two ATC classes differ as well as GL clusters. The significance level of comparing drugs in ATC classes/GL clusters against each other was adjusted using a rigorous paired Wilcoxon signed-rank test with Bonferroni correction to control family-wise type I error(31).”

3. Please clarify if clustering was done following dimensionality reduction or whether these were separate steps.

Thank you for your comment. The clustering referred to the grouping of nodes in Circos plots (Figure 2-B). We have added an explanation in the main text from Lines 424-427:

“Figure 3A shows the association between the remaining 22 drugs after then the elimination process from the penalized regression GLASSO. It means that the clustering was done after applying the GLASSO method which is a dimensionality reduction method.”

The bridges between any drugs indicated the clustering. The data used were obtained from the EBGM scores of 22 drugs selected by the GLASSO method.

4. The use of buzz words like "machine learning" and "AI" is over done given the preponderance of work is based in traditional statistics. I would advise removing machine learning and AI from the title as it sort builds up an appearance that does not hold when read by a machine learning audience. A "data driven approach" is appropriate and best represents the work.

We have changed AI to data-driven in the related main text. But the fact is that we used data mining and data analytics as parts of artificial intelligence and machine learning for the extraction of messy, incomplete, implicit, or previously unknown information for potentially useful data in all of the places in the work. EBGM and GLASSO involve statistical learning. Data mining techniques are the core procedure we have applied to other parts of this study. That’s why we referred to it as “machine learning” and “AI”. However, we have changed the title to be more specific in our title:

“Data-driven methodology for discovery and response to pulmonary symptomology in hypertension through statistical learning and data mining: Application to COVID-19 related pharmacovigilance”.

5. Where possible, remove first person in Results and Discussion.

We have changed almost all the first person to third person in Results and Discussion.

6. Title: "discovery and response to pulmonary symptomology in hypertension". It is confusing (at least) to me what the authors want to say. For example, does "discovery" here refer to "diagnosis"?

“Discovery and response” in the title refer to the detection and identification of ADEs in combination with worse drugs after applying machine learning and data mining techniques.

7. Abstract: The one sentence in Methods is not clear and specific enough. In addition, it is hard to follow how the results derive to the conclusions.

Yes, that’s correct. This is due to the limit of words for the abstract, we can only partially explain our method. Below is the edited part of the abstract:

“Data from patients with hypertension were retrieved and integrated from the FDA Adverse Event Reporting System. 134 antihypertensive drugs out of 1151 drugs were filtered and then evaluated using the Empirical Bayes Geometric Mean (EBGM) of the posterior distribution to build ADE-drug profiles with an emphasis on the pulmonary ADEs (pADE). Afterward, the Graphical Least Absolute Shrinkage and Selection Operator (GLASSO) captured drug associations based on pADEs by correcting hidden factors and confounder misclassification. Selected drugs were then compared using the Friedman test in drug classes and clusters obtained from GLASSO.” (Lines 31-38)

8. Line 51: typo: indidvual -> individual

Thank you for your comment. We have corrected the typo.

9. Lines 130-168: I would suggest to move the lists of the ADEs names (e.g., lines 152-165) into a table or a supplementary material.

It is a great suggestion. We have moved the ADEs to a new table in the Supporting Information file, Supplementary file 1, Line 310.

10 Line 176: Please add a reference supporting the cut-off criterion (RR >2).

Thank you for your comment. We have added some references supporting the cut-off criterion. Applying a threshold of interest, i.e., a twofold RR, to perform a relevant effect estimate has been reported in previous studies:

[1] Clayton DG. Bayesian methods for mapping disease risk. Geographical and environmental epidemiology: methods for small-area studies. 1992:205-20.

[2] Richardson S, Thomson A, Best N, Elliott P. Interpreting posterior relative risk estimates in disease-mapping studies. Environmental health perspectives. 2004 Jun;112(9):1016-25.

A study toxic tort litigation directly used RR > 2 as the threshold for relationship

[3] Carruth RS, Goldstein BD. Relative risk greater than two in proof of causation in toxic tort litigation. Jurimetrics. 2001 Jan 1:195-209.

Other studies claim that RR of 2 is the cutoff point to define the association (a disease to a therapy):

[4] Balkau B, Bertrais S, Ducimetiere P, Eschwege E. Is there a glycemic threshold for mortality risk?. Diabetes care. 1999 May 1;22(5):696-9.;

[5] Curtis RE, Boice Jr JD, Stovall M, Bernstein L, Greenberg RS, Flannery JT, Schwartz

AG, Weyer P, Moloney WC, Hoover RN. Risk of leukemia after chemotherapy and radiation treatment for breast cancer. New England Journal of Medicine. 1992 Jun 25;326(26):1745-51.

It is worth mentioning that the purpose of having RR > 2 in our study was to eliminate the noise-driven reports in order to perform a comparative analysis on the most influential drugs.

Here is what we have edited in the main text (Lines 322-325):

“It is worth mentioning that several studies have reported RR > 1.5 or 2, or a particular threshold larger than 1 to justify the association with more confidence, especially in the presence of additive noise with the unidentified distribution (32-34).”

11. Figure 1: It is hard to follow how to interpret the results. Also, please increase the font size in the figure.

We appreciate the comments and suggestions on figures and corresponding context. First we should mention that the order of figures change and Figure 1 is now shown by Figure 2. To respond to the comments raised by the reviewers, we have changed the font size in Figure 2-A, Figure 2 A and B have been placed on two separate pages, and the paragraph linked to Figure 2 has been updated to become more legible. However, we would like to mention here that the purpose of Figure 2-A was to demonstrate how messy the distribution of 134 drugs and their loadings using a well-known statistical process, even in a 3D plot, and to draw the audience’s attention to our tactic of implementing big data analysis. Also, we mentioned in the caption that

“Click the following URL to see the figure: https://1data.life/pages/publication/figure1B.html”